# Improved diagnosis of extrapulmonary tuberculosis in adults with and without HIV in Mbeya, Tanzania using the MPT64 antigen detection test

Erlend Grønningen[1,2]*, Marywinnie Nanyaro[3], Lisbet Sviland[4,5], Esther Ngadaya[3], William Muller[3,6], Lisete Torres[3,6], Sayoki Mfinanga[3], Tehmina Mustafa[1,2]

1 Department of Global Public Health and Primary Care, Centre for International Health, University of Bergen, Bergen, Vestland, Norway, 2 Department of Thoracic Medicine, Haukeland University Hospital, Bergen, Vestland, Norway, 3 National Institute for Medical Research, Dar es Salaam, The United Republic of Tanzania, 4 Faculty of Medicine, Department of Clinical Medicine, University of Bergen, Bergen, Vestland, Norway, 5 Department of Pathology, Haukeland University Hospital, Bergen, Vestland, Norway, 6 Mbeya Zonal Referral Hospital, Mbeya, The United Republic of Tanzania

* erlend.gronningen@uib.no

**Data Availability Statement:** All data used in this study is available in the supplementary information.

## Abstract

Extrapulmonary tuberculosis (EPTB) in People Living with HIV (PLWHIV) is a diagnostic challenge. Our immunochemistry based MPT64 antigen detection test has shown improved sensitivity compared to current laboratory tests in the resource limited diagnostic setting. The aim of this study was to validate the implementability and diagnostic performance of the test in PLWHIV and HIV negative adults in a HIV endemic Tanzanian setting. Adult (>18 y) presumptive EPTB patients were prospectively enrolled at Mbeya Zonal Referral Hospital and followed to the end of treatment or until an alternative diagnosis was reached. Suspected sites of infection were sampled and were subject to routine diagnostics, GeneXpert MTB/RIF assay and the MPT64 test. The performance of the diagnostics tests was assessed using a composite reference standard that included clinical suspicion, mycobacterial culture, response to anti-tuberculosis (TB) therapy, cytological and radiological findings. Patients (N = 168) were categorized as 21 confirmed TB, 23 probable TB and 44 possible TB cases, 69 patients were categorized as non-TB cases and 11 were uncategorized. In the TB group, the three most common infections were adenitis (41%), peritonitis (19%) and pleuritis (14%). The TB and non-TB groups did not differ in HIV seropositivity (46% vs 42%) Among HIV negative and PLWHIV, the MPT64 test had a sensitivity of (91% vs 78%), specificity (75% vs 86%), positive predictive value (80% vs 88%), negative predictive value (89% vs 74%), and accuracy (84% vs 81%), respectively. Performance was not significantly reduced in PLWHIV, and sensitivity was higher than in the currently used tests, including the GeneXpert MTB/RIF assay. The MPT64 test improved the diagnosis of EPTB, irrespective of HIV status. The test performed better than currently used diagnostic test. The test was implementable in a tertiary level hospital with basic pathology services in a HIV endemic Tanzanian setting.

**Funding:** This work was partly supported by the Research Council of Norway through the Global Health and Vaccination Programme [project number 234457]. This project is part of the EDCTP2 programme supported by the European Union. The first author (EG) received a grant from the Norwegian Medical Association to cover expenses relating to travels and the gathering of data in Tanzania. The funders had no role in study design, data collection and analysis, decision to publish, or preparation of the manuscript.

**Competing interests:** The authors have declared that no competing interests exist.

## Introduction

Co-infection with Human Immunodeficiency Virus (HIV) and *Mycobacterium tuberculosis* (Mtb) is still a cause of significant mortality and morbidity. Of the 9.9 million that fell ill with tuberculosis (TB) globally in 2020, 8% were also infected with HIV [1]. TB remains the leading cause of death among people living with HIV (PLWHIV), and the World Health Organization (WHO) estimates that 214.000 HIV infected patients died of TB in 2020. Worldwide 37.7 million people are living with HIV [2]. HIV is associated with both pulmonary TB (PTB) and extrapulmonary TB (EPTB). With reducing CD4 counts the proportion of EPTB cases relative to PTB increases [3]. Previous studies have shown that those who are co-infected with HIV and EPTB are diagnosed later and have poorer prognosis compared to EPTB patients without HIV [4]. EPTB has been considered an AIDS-defining condition [5], but it has not been given priority as it is less transmissible than PTB and thus less of a public health threat [6].

EPTB is a diagnostic challenge, and a high clinical suspicion is required due to the non-specific clinical presentation. Current diagnostic tests often fail to detect the disease because of the paucibacillary nature of EPTB [7]. In PLWHIV the diagnosis of EPTB is even more challenging as the clinical presentation and histopathology/cytology features change. The current gold standard test, Mtb culture, has a moderate sensitivity on extrapulmonary samples, but is still necessary for drug susceptibility testing [3]. WHO has recommended the use of GeneXpert MTB/RIF assay (GeneXpert, Cepheid, Sunnyvale, California, USA) for EPTB samples [8], and a recent Cochrane review found excellent specificity, but varying sensitivity across different specimens, comparable to Mtb culture [9]. Acid fast bacilli (AFB) staining is not sensitive enough to serve as a stand-alone test for EPTB [10,11]. Fine Needle Aspiration Cytology (FNAC) and histology of mass lesions has a high sensitivity, but sensitivity and specificity are reduced in advanced HIV disease. Differentiation from other granulomatous conditions is also a challenge. Serological test and IFN-γ release assays lack accuracy to identify the progressors from infection to TB disease and are not suitable to detect EPTB [12].

Immunostaining with an anti-MPT64 antibody has shown excellent sensitivity and specificity when compared to nested PCR, performs equally well in HIV coinfection and in the routine diagnostic setting using cytology/FNAC [7,10,13–19]. MPT64 is a protein that is secreted by Mtb and is not detected in non-tuberculous mycobacteria or Bacillus Calmette-Guerin strains with RD2 deletion [20]. The MPT64 antigen detection test (MPT64 test) has shown a sensitivity of 81% and a specificity of 100% on pleural biopsies for TB pleuritis in HIV-coinfected patients [15]. In two recently published studies on the performance of the MPT64 test in routine diagnostic settings in Tanzania, the MPT64 test had a sensitivity of 92% when compared to a composite reference standard (CRS) for pediatric EPTB with various presentations in Mbeya [17], a 100% sensitivity in pediatric TB adenitis and 65% in all EPTB patients in Zanzibar [16]. Performance of the test in an adult cohort with a high prevalence of HIV has not been validated in a real-life routine diagnostic setting.

Tanzania is one of the 30 TB high burden countries and has an overlapping epidemic with HIV [1,2]. In 2018, 28% of all new TB cases were HIV infected. There were 75,828 notified cases of TB in 2018, of which 20% were extrapulmonary [21]. Mbeya Zonal Referral Hospital is a tertiary level hospital with existing histopathology services that serves approximately 8 million people in six regions in the Southern Highlands in Tanzania. Mbeya region has an adult (>15 years) HIV prevalence of 9,3% [22].

The aims of this study were to (i) assess if the MP64 test was implementable in a tertiary level hospital in a HIV endemic, low-income country with a high TB burden, and (ii) to validate the performance of the test against currently used diagnostic tests, including GeneXpert, in HIV infected adults with suspected EPTB.

## Materials and methods

### Study population and inclusion of patients

This study was part of an open prospective cohort study in Mbeya Zonal Referral Hospital (MZRH) conducted between April 1st, 2016—July 31st, 2017, to validate the MPT64 test [17]. A cohort of adult patients (>18 years old) was nested to assess the performance of the test in a cohort with a high HIV prevalence. From both the out- and inpatient services clinicians were asked to recruit patients with presumed EPTB. Clinicians did not receive trainings or guidance in when to suspect EPTB but followed standard procedures. The study design is shown in Fig 1.

All patients gave their informed consent to be included in the study. Patients who were on anti-TB therapy on inclusion or had received anti-TB therapy during the last year were excluded. Patients that were unclassifiable per our CRS were also excluded.

Collection of data and clinical assessment of the patients were done by medical officers hired for the purpose of the study. Additional clinical information, radiological examinations and laboratory investigations were retrieved from the electronic hospital records for the study participants.

On inclusion patients were asked to disclose their HIV serostatus and if they were on anti-retroviral therapy (ART). Patients with unknown HIV serostatus were not routinely tested but rapid diagnostic tests were used on clinicians request per national guidelines [23]. Measurement of CD4 counts and assessment of viral loads were performed per request of the clinicians and not routinely.

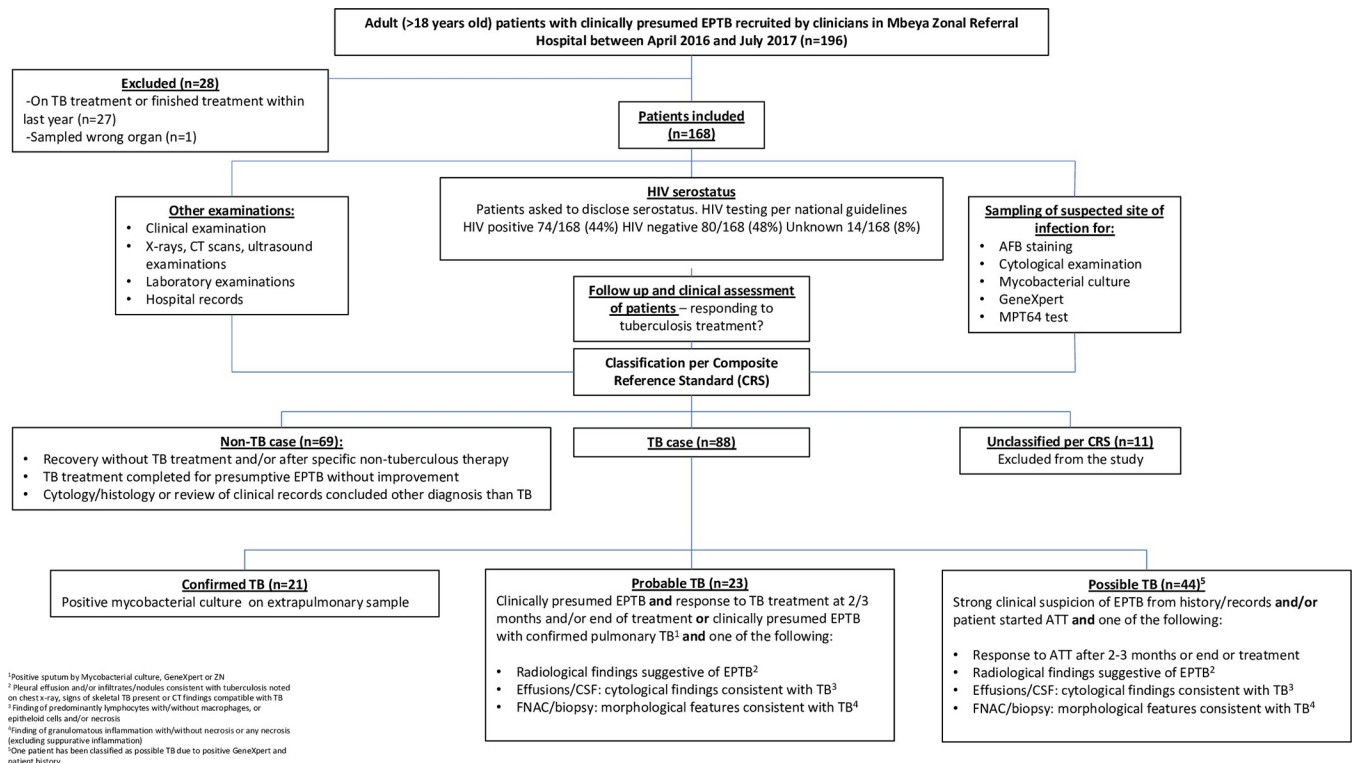

**Fig 1. Study design, follow up and patient categorization.**

Follow ups of the patients were performed 2–3 months after inclusion and at the end of anti-TB treatment. Follow ups were discontinued if another diagnosis than EPTB was reached. Additional tracings of patients were done if they had been marked as lost to follow up.

### Definitions of patients

Patients were classified according to a CRS into TB and non-TB groups, as shown in Fig 1. The CRS included clinical suspicion, mycobacterial culture, concomitant confirmed PTB, response to anti-TB treatment, radiological findings, cytology findings compatible with EPTB in effusions and morphological features on FNAC. The TB group was further subclassified to account for the degree of uncertainty of the diagnosis; confirmed TB, probable TB, and possible TB. The criteria for each category is shown in Fig 1. If there was ambiguity of the classification of patients after the initial classification by EG and MN, a second classification was done by TM. The second classification had all information available other than the MPT64 test result and previous classification.

### Diagnostic samples

The suspected site of infection was sampled by the pathologists (LT and WM). While applying minimal negative pressure, a 22-gauge needle attached to a syringe was used to aspirate material from superficial, palpable lesions and enlarged lymph nodes. The material was then spread out on to glass slides for cytological evaluation, MPT64 staining and Ziehl-Neelsen staining (ZN). The needle and the syringe were rinsed using sterile saline to obtain material for Mtb culture and GeneXpert.

### Diagnostic tests

ZN was used for microcopy for AFB by the pathologists. Samples of Mtb culture and GeneXpert were analyzed at the Central Tuberculosis Reference Laboratory at Muhimbili National Hospital in Dar es Salaam. Mtb culture was performed on Lowenstein-Jensen medium, and both Mtb culture and GeneXpert was performed according to the protocol at the Central Tuberculosis Reference Laboratory, in accordance with the WHO protocol [24].

The Papanicolaou stain was used for staining cytological slides, while haematoxylin and eosin was used for histological slides. Finding of granulomatous inflammation with/without necrosis or any necrosis (excluding suppurative inflammation) on FNAC was defined as compatible with TB [25]. The same findings were also defined as compatible with TB on biopsy. Cytological findings in effusions compatible with TB were defined as composed of predominantly lymphocytes and/or macrophages, or epithelioid cells and/or necrosis [26–29].

The two pathologists in MZRH (LT and WM) received training in evaluating immunocytochemistry/immunohistochemistry (immunostaining), and two technicians were trained in the procedure of immunostaining. In short, through decreasing grades of alcohol the previously alcohol fixed smears were hydrated and washed in distilled water. To inhibit endogenous peroxidase activity the smears were incubated with hydrogen peroxide. The techniques of staining with the anti-MPT64 antibody have been described in detail previously and was repeated in this study [17], but with some modification [16]. To demonstrate the MPT64 antigens we used an in-house polyclonal anti-MPT64 antibody at 1/250 dilution and Dako kit (Dako Envision + System -HRP, K4009, Dako, Glostrup, Denmark). The local pathologists (WM and LT) assessed the immunostained slides while being blinded for the results of Mtb culture and GeneXpert. If reddish granular intracytoplasmic staining or extracellular staining was found in necrotic areas, the immunostaining was regarded as positive, as shown in Fig 2.

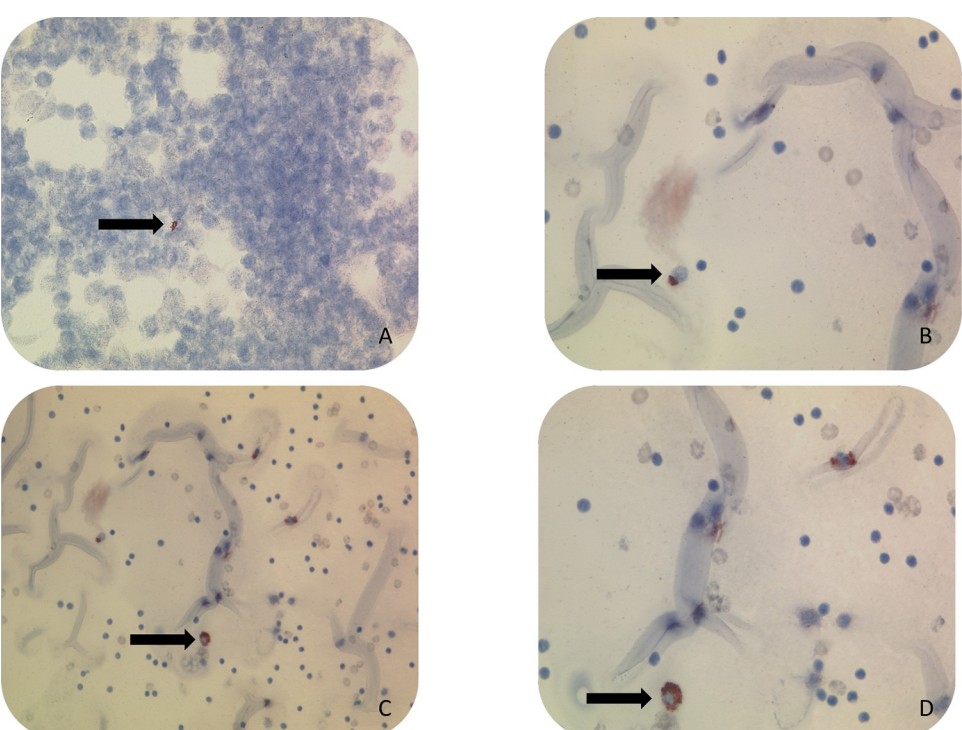

**Fig 2. Immunostaining of cytological smears showing positive MPT64 staining as brown/red granular staining within macrophages (arrows).** A: Lymph node (X40). The slide does not show clearly if the staining was inside a macrophage or in the background due to rupture of the macrophage. B: Cytological smear of pleural effusion showing positive signal for MPT64 (brown-red color) inside a macrophage (X40). C: Cytological smear of pleural effusion showing positive signal for MPT64 (brown-red color) inside macrophages (X20). D: Cytological smear of Pleural effusion showing positive signal for MPT64 (brown-red color) inside macrophages (X40).

## Quality control of immunostaining

To assess the quality of the reading of the MPT64 test an independent reading of 20 randomly selected immunostained slides was performed by a pathologist from a Norwegian university hospital (LS). Agreement was found on 12/20 (60%) MPT64 slides (55% positive, 5% inconclusive). 8/20 (40%) had a discordant result. Among the discordant cases 6 (30%) were labelled as inconclusive by LS, 1 (5%) positive by the local pathologist and negative by LS and 1 (5%) positive by LS and inconclusive by the local pathologist.

## Statistical analysis

Statistical Package for the Social Sciences (SPSS) for Mac, version 27.0, was used for statistical analysis. Differences in categorical variables were assessed using Chi square test or Fishers exact test (when cells had a count below 5). Our data was not normally distributed, hence non-parametric tests were used for group comparisons. A p-value < .05 was defined as statistically significant. Cross-tabulation was used for calculating sensitivity, specificity, positive predictive value (PPV), negative predictive value (NPV) and accuracy.

## Ethical considerations

The regional Committee for Medical Research Ethics in Norway, REK Helse-Vest (2014/46/REK vest), and the Ethical committee for biomedical research at the National Medical Research Coordinating Committee in Tanzania (NIMR/HQ/R.8a/Vol.IX/2142) approved the

project. All study participants were given information about the study and gave written consent. Relevant guidelines and regulations were followed when performing the experiments. No invasive sampling was performed primarily for study purposes, but per request of the clinicians in MZRH based on patient management. HIV testing was performed per national guidelines, and not primarily for the study. Additional information regarding the ethical, cultural, and scientific considerations specific to inclusivity in global research is included in the Supporting Information (S1 Text).

## Results

### Study population

A total of 168 cases were included in the study. Eleven cases were unclassifiable according to the CRS and were excluded. Among the 157 cases, 88 (56%) were classified as TB cases, and 69 (44%) as non-TB cases, as shown in Fig 1. Of the 88 TB cases, 21 (24%) were categorized as confirmed TB (positive Mtb culture), 23 (26%) as probable TB and 44 (50%) as possible TB. The TB and non-TB groups did not differ in HIV seropositivity (46% vs 42%), or in percentage receiving ART (93% vs 90%), as shown in Table 1. No statistically significant difference was found in baseline or current CD4 counts. However, only 17/69 (25%) PLWHIV had current CD4 counts available (TB group N = 10, mean = 206, non-TB group N = 7, mean = 307, p = .151). Baseline CD4 counts were available in 13/69 (19%) (TB group N = 9, mean = 317/ median = 246, non-TB group N = 4, mean = 311/median = 157, p = .918).

Most patients included in the study were male (66%), but the TB and non-TB groups did not differ in gender distribution. The TB group was significantly younger than the non-TB group (median 36 vs 42, p = .003). Most patients in both the TB and non-TB groups were recruited from the inpatient department. The three most common suspected sites of infection were TB adenitis (41% vs 28%, TB group vs non-TB group, respectively), pleuritis (19% vs 28%) and peritonitis (14% vs 23%). A statistically significant higher percentage of non-TB cases (20% vs 7%, p = .012) reported to have received anti-TB therapy in the past.

### Performance of diagnostic tests

**MPT64 test.** The MPT64 test had an overall sensitivity of 87% on 75 TB samples compared to the CRS, as shown in Tables 2 and 3. The sensitivity of the MPT64 test was not significantly affected by HIV serostatus (78% vs 91%, HIV + vs HIV-, respectively, p = 0.175). The test had an overall specificity of 82% with a total of 8 false positive samples in the non-TB group, of which 3 in HIV negative and 5 in PLWHIV, as shown in Table 2. The overall specificity of the test was not significantly affected by serostatus (86% vs 75%, HIV+ vs HIV-, p = 0.454). The test had an overall PPV, NPV and accuracy of 86%, 83% and 85%, respectively.

The MPT64 test had an overall sensitivity of 95% on 37 FNA samples (all organs), as shown in Tables 2 and 3. The sensitivity of the test on FNA was not affected by HIV serostatus (93% vs 94%, HIV+ vs HIV-, p = 1). Specificity was 84% with 3 false positive samples: 1 benign tumor and 2 cases of mastitis.

In effusions the MPT64 test had an overall sensitivity of 79% on 38 samples, as shown in Tables 2 and 3. The difference in sensitivity of the test between HIV infected and HIV negative was not statistically significant (65% vs 90%, HIV+ vs HIV-, respectively, p = 0.113).

The higher sensitivity of the MPT64 test on FNA than in effusions was found not to be statistically significant (95% vs 79%, FNA vs effusions, respectively, p = 0.086), but results need to be interpreted with caution as the sample size is small.

**Ziehl-Neelsen staining.** ZN staining had an overall sensitivity of 10% on 61 samples compared to the CRS, as shown in Table 3. The test had a perfect specificity of 100% and a

**Table 1. Baseline characteristics of study participants grouped as TB and non-TB.**

| | TB (n = 88) | Non-TB (n = 69) | P-value TB/non-TB |
|---|---|---|---|
| **Patient group** | | | |
| Total (n = 157) | 88/157 (56%) | 69/157 /44%) | |
| **Serostatus** | | | |
| HIV-infected | 40/88 (46%) | 29/69 (42%) | .527 |
| HIV negative | 39/88 (44%) | 35/69 (51%) | |
| Unknown [2] | 9 (10%) | 5/69 (7%) | |
| **ART among HIV-infected [3]** | | | |
| Yes | 37/40 (93%) | 26/29 (90%) | .956 |
| No | 3/40 (7%) | 2/29 (7%) | |
| Unknown [2] | 0 | 1/29 (3%) | |
| **Sex** | | | |
| Female (n = 54) | 34/88 (39%) | 20/69 (29%) | .137 |
| Male (n = 103) | 54/88(61%) | 49/69 (71%) | |
| **Age years (Median—Range)** | 36 (19–75) | 42 (20–85) | .003 [1] |
| **Department** | | | |
| Inpatient (n = 100) | 52/88 (59%) | 48/69 (70%) | .176 |
| Outpatient (n = 57) | 36/88 (41%) | 21/69 (30%) | |
| **Suspected site of infection** | | | |
| Adenitis | 36/88 (41%) | 19/69 (28%) | |
| Peritonitis | 17/88 (19%) | 19/69 (28%) | |
| Pleuritis | 12/88 (14%) | 16/69 (23%) | |
| Multiorgan involvement [4] | 11/88 (13%) | 7/69 (10%) | |
| PTB with EPTB | 8/88 (9%) | 2/69 (3%) | |
| Mastitis | 1/88 (1%) | 3/69 (4%) | |
| Other [5] | 3/88 (3%) | 3/69 (4%) | |
| **Self-reported previous TB treatment (>1y prior to enrollment)** | | | |
| Yes (n = 20) | 6/88 (7%) | 14/67 (20%) | .012 |
| No (n = 137) | 82/88 (93%) | 54/67 (80%) | |

[1] Mann whitney test. Other p-values chi square.

[2] Unknown HIV status removed from p-value calculation. Unkown ART use removed from p-value.

[3] Yes: 55 reports to be on ART on inclusion, 4 starts ART after initiating ATT, 4 newly diagnosed that starts ART; No: 3 reports not to be on ART, 2 newly diagnosed and dies before ART initiation.

[4] Includes disseminated TB (2 patients in TB group, 2 non-TB group) and patients with suspected EPTB in 2 sites (9 patients in TB group, 5 non-TB group).

[5] Includes: TB group, 1 case of: Abdominal cyst, ovarial, thyroid. Non-TB group, 1 case of: Scrotum, skin, liver.

correspondingly high PPV, NPV and accuracy was lower at 47% and 49%, respectively. HIV serostatus did not affect the overall sensitivity of the test (15% vs 7%, HIV+ vs HIV-, respectively, p = 0.42). The sensitivity of ZN staining was better on FNA (19%, 6/32) as compared to the effusions (0%, 0/29).

**Mycobacterial culture.** Mtb culture had an overall sensitivity of 28% on 75 samples compared to the CRS, as shown in Table 3. HIV serostatus affected the overall sensitivity of the test (17% vs 42%, HIV+ vs HIV-, respectively, p = 0.018) with significantly poorer sensitivity in the

**Table 2. Positive MPT64 test results across different specimens among HIV+/HIV- study participants.** All TB/non-TB rows includes participants with unknown HIV status.

| | All TB patients (n = 88) | HIV + (n = 40) | HIV- (n = 39) | |
|---|---|---|---|---|
| TB samples | MPT64 (n = 75) [1] | MPT64 (n = 32) [1] | MPT64 (n = 35) [1] | p-value difference sensitivity HIV+/HIV- |
| All samples (n = 88) | 65/75 (87%) | 25/32 (78%) | 32/35 (91%) | 0.175 |
| Lymph node FNA (n = 39) | 30/32 (94%) | 12/13 (92%) | 14/15 (93%) | |
| FNA other organ (n = 5) [3] | 5/5 (100%) | 2/2 (100%) | 1/1 (100%) | |
| All FNA (n = 44) | 35/37 (95%) | 14/15 (93%) | 15/16 (94%) | 1 |
| Pleural effusion (n = 23) | 13/18 (72%) | 8/11 (73%) | 4/6 (67%) | |
| Ascites (n = 20) | 17/20 (85%) | 3/6 (50%) | 13/13 (100%) | |
| All effusions (n = 44) | 30/38 (79%) | 11/17 (65%) | 17/19 (90%) | 0.113 |
| | **All non-TB patients (n = 69)** | **HIV+(n = 29)** | **HIV- (n = 35)** | |
| Non- TB samples | MPT64 (n = 45) 2 | MPT64 (n = 21) [2] | MPT64 (n = 20) [2] | p-value difference specificity HIV+/HIV- |
| All samples (n = 69) | 8/45 (18%) | 3/21 (14%) | 5/20 (25%) | 0.454 |
| Lymph node FNA (n = 22) | 1/17 (6%) | 1/9 (11%) | 0/5 (0%) | |
| FNA other organ (n = 6) [4] | 2/2 (100%) | 0 | 2/2 (100%) | |
| All FNA (n = 28) | 3/19 (16%) | 1/9 (11%) | 2/7 (29%) | 0.55 |
| Pleural effusion (n = 20) | 1/15 (7%) | 1/8 (13%) | 0/6 (0%) | |
| Ascites (n = 21) | 4/11 (36%) | ¼ (25%) | 3/7 (43%) | |
| All effusions (n = 41) | 5/26 (19%) | 2/12 (17%) | 3/13 (23%) | 1 |

[1] MPT64 samples: All: 7 uncertain, 6 missing, HIV+: 2 missing samples, 6 uncertain, HIV-: 4 missing samples.

[2] MPT64 samples: All: 3 uncertain, 21 missing, HIV+: 1 uncertain, 7 missing, HIV-: 1 uncertain, 14 missing.

[3] FNA other organ TB group 2 cyst in abdomen, 1 thyroid, 1 ovaries, 1 mastitis.

[4] FNA other organ non-TB group: 3 breast, 1 scrotum, 1 liver, 1 skin.

HIV infected. The test had a perfect specificity of 100%, but due to the poor sensitivity the test had an overall accuracy of 60% and a NPV of 52%. On FNA (all organs) Mtb culture had an overall sensitivity of 34% (13/38), as shown in Table 3. The sensitivity of the test on FNA was not significantly influenced by HIV serostatus (28% vs 47%, HIV+ vs HIV-, respectively, p = 0.261). In effusions Mtb culture had an overall sensitivity of 22% (8/37), and the test sensitivity on effusions was significantly lower in HIV infected patients (6% vs 39%, HIV+ vs HIV-, respectively, p = 0.04.).

**GeneXpert.** GeneXpert had an overall sensitivity of 33% on 87 samples compared to the CRS, as shown in Table 3. HIV serostatus affected the overall sensitivity significantly, with a poorer performance among HIV infected (23% vs 44%, HIV+ vs HIV-, respectively, p = 0.046). The test had an overall specificity of 98% as one HIV negative case was false positive. The PPV was 97%, but with a lower accuracy at 62% and NPV at 54%. On FNA (all organs) the test had an overall sensitivity of 39% (17/44), as shown in Table 3. The sensitivity of GeneXpert on FNA was not significantly influenced by HIV serostatus (35% vs 44%, HIV + vs HIV-, respectively, p = 0.552). GeneXpert had an overall sensitivity of 28% (12/43) in effusions, and HIV serostatus significantly affected the sensitivity of GeneXpert (10% vs 43%, HIV + vs HIV-, respectively, p = 0.033).

The test results for ZN, Mtb culture and GeneXpert across different samples is shown in S1 Table.

**Comparison of diagnostic tests.** Positive Mtb culture and GeneXpert results and their correlaton to the MPT64 test is shown in Fig 3. Regardless of serostatus there was perfect correlation between a positive Mtb culture result and a positive MPT64 test, however the correlation was not inverse. In total 44 tests were positive on the MPT64 test, but negative on Mtb

**Table 3. Validation of all diagnostic tests by using composite reference standard and culture as the reference standard according to the HIV serostatus.**

| All patients—CRS as reference1 (n = number of samples)[1] | Sensitivity (95% C.I) | Specificity (95% C.I) | PPV (95% C.I) | NPV (95% C.I) | Accuracy (95% C.I) |
|---|---|---|---|---|---|
| ZN(N, TB = 61, non-TB = 48) | 10% (4–20%) | 100% (93–100%) | 100% | 47% (44–49%) | 49% (40–59%) |
| Mtb Culture(N, TB = 75, non-TB = 60) | 28% (18–40%) | 100% (94–100%) | 100% | 52% (49–56%) | 60% (51–68%) |
| GeneXpert(N, TB = 87, non-TB = 66) | 33% (24–44%) | 98% (92–100%) | 97% (80–100%) | 54% (50–57%) | 62% (54–70%) |
| MPT64 (N, TB = 75, non-TB = 45) | 87% (77–93%) | 82% (68–92%) | 86% (77–92%) | 83% (73–90%) | 85% (77–91%) |
| **HIV-infected[2]** | | | | | |
| ZN (N, TB = 27, non-TB = 21) | 15% (4–34%) | 100% (84–100%) | 100% | 46% (42–50%) | 51% (36–65%) |
| Mtb Culture (N, TB = 36, non-TB = 25) | 17% (6–33%) | 100% (86–100%) | 100% | 46% (43–50%) | 52% (39–65%) |
| GeneXpert (N, TB = 40, non-TB = 28) | 23% (11–38%) | 100% (88–100%) | 100% | 48% (44–52%) | 55% (43–67%) |
| MPT64 (N, TB = 32, non-TB = 21) | 78% (60–91%) | 86% (64–97%) | 88% (72–96%) | 74% (59–85%) | 81% (68–91%) |
| **HIV uninfected [3]** | | | | | |
| ZN (N, TB = 29, non-TB = 23) | 7% (1–24%) | 100% (85–100%) | 100% | 49% (46–51%) | 51% (36–65%) |
| Mtb Culture(N, TB = 33, non-TB = 30) | 42% (25–61%) | 100% (88–100%) | 100% | 61% (53–67%) | 69% (57–80%) |
| GeneXpert(N, TB = 39, non-TB = 33) | 44% (28–60%) | 97% (84–100%) | 94% (70–99%) | 60% (53–67%) | 69% (57–79%) |
| MPT64 (N, TB = 35, non-TB = 20) | 91% (77–98%) | 75% (51–91%) | 80% (66–90%) | 89% (72–96%) | 84% (71–92%) |
| **All FNA samples—CRS as reference [4]** | | | | | |
| ZN(N, TB = 32, non-TB = 19) | 19% (7–36%) | 100% (82–100%) | 100% | 44% (40–48%) | 50% (36–65%) |
| Mtb Culture (N, TB = 38, non-TB = 24) | 34% (19–51%) | 100% (86–100%) | 100% | 49% (44–55%) | 60% (47–72%) |
| GeneXpert(N, TB = 44, non-TB = 28) | 39% (24–55%) | 100% (88–100%) | 100% | 51% (45–57%) | 63% (50–74%) |
| MPT64 (N, TB = 37, non-TB = 19) | 95% (82–99%) | 84% (60–97%) | 90% (77–96%) | 91% (72–97%) | 91% (80–97%) |
| **All effusion sample—CRS as reference [5]** | | | | | |
| ZN(N, TB = 29, non-TB = 29) | 0% (0–12%) | 100% (88–100%) | - | 48% (48–48%) | 48% (35–62%) |
| Mtb Culture (N, TB = 37, non-TB = 36) | 22% (10–38%) | 100% (90–100%) | 100% | 54% (50–58%) | 59% (47–71%) |
| GeneXpert(N, TB = 43, non-TB = 38) | 28% (15–44%) | 97% (86–100%) | 92% (61–99%) | 56% (51–60%) | 61% (50–72%) |
| MPT64 (N, TB = 75, non-TB = 45) | 79% (63–90%) | 81% (61–93%) | 82% (66–91%) | 78% (65–87%) | 80% (68–89%) |
| **All patients—culture defined TB [6]** | | | | | |
| ZN(N, TB = 20, non-TB = 77) | 15% (3–38%) | 96% (89–99%) | 41% (13–76%) | 86% (84–88) | 83% (75–90%) |
| GeneXpert(N, TB = 21, non-TB = 114) | 95% (76–100%) | 95% (89–98%) | 77% (60–88%) | 99% (94–100%) | 95% (90–98%) |
| MPT64 (N, TB = 19, non-TB = 83) | 100% (82–100%) | 47% (36–58%) | 26% (22–30%) | 100% | 55% (45–65%) |

[1]Disease prevalence set to 88/157. EPTB patients defined by composite reference standard.

[2]Disease prevalence set to 40/69. EPTB patients defined by composite reference standard.

[3]Disease prevalence set to 39/74. EPTB patients defined by composite reference standard.

[4]Disease prevalence set to 44/72. EPTB patients defined by composite reference standard.

[5]Disease prevalence set to 44/85. EPTB patients defined by composite reference standard.

[6]Disease prevalence set to 21/135. EPTB is defined as culture positive. Culture negatives added to non-TB group.

culture. The correlation with GeneXpert was perfect for 26 GeneXpert positive samples, but one false positive GeneXpert result had a corresponding negative MPT64 test. 45 samples had a positive MPT64 result, but a correspondingly negative GeneXpert.

Using Mtb culture as the reference standard, the MPT64 test had a sensitivity of 100% and a specificity of 47%, as shown in Table 3.

**FNAC and cytology and correlation with MPT64 results.** Table 4 shows correlation of cytology and FNAC results with MPT64 test results. The correlation of ZN, Mtb culture and GeneXpert with FNAC and cytology findings is found in S2 and S3 Tables.

FNAC results were available for 44 EPTB patients. With the FNAC criteria for TB used in this study, FNAC showed a sensitivity of 86% on 44 samples, of which 39 were from lymph nodes. MPT64 results were available for 37 of the patients that also had FNAC performed.

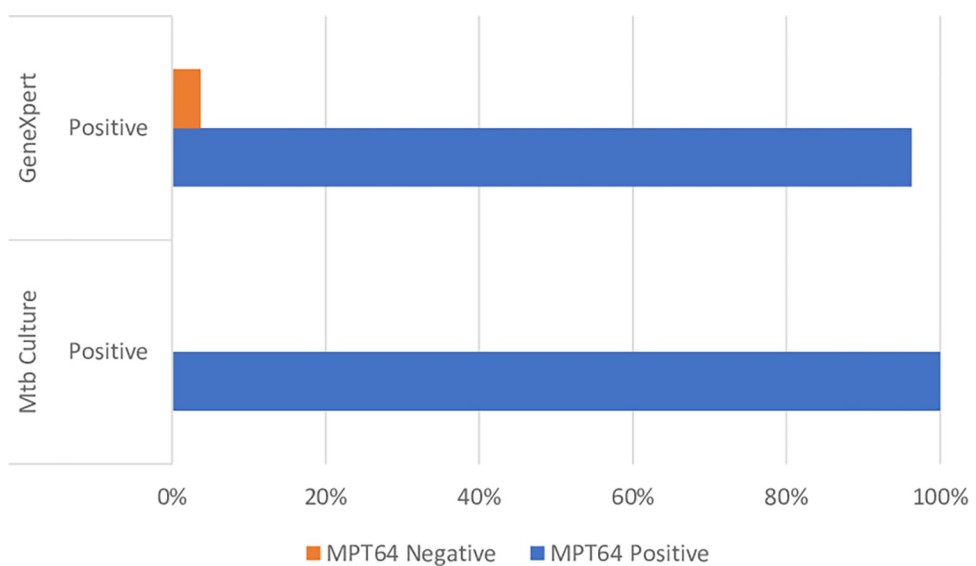

**Fig 3. Positive mycobacterial culture/GeneXpert and correlation to MPT64 test result.**

MPT64 was positive in 95% (18/19) samples showing granulomatous inflammation +/- necrosis, and 100% (12/12) samples showing mixed inflammatory cells with necrosis. Only one sample that showed typical features of TB on FNAC had a negative MPT64 test. Of the TB cases not displaying typical findings on FNAC, MPT64 was positive in 5/6 (83%). Combining positive FNAC and MPT64 results would correctly have diagnosed 97% (36/37) of the cases that

**Table 4. Positive MPT64 test results and their correlation to findings on FNAC and cytological findings in effusions (n = number of MPT64 samples).**

| Findings compatible with EPTB in bold | | | | |
|---|---|---|---|---|
| | **All EPTB (n = 37)1** | **HIV + (n = 15)** | **HIV- (n = 16)** | **Non-TB (n = 19)** |
| Total positive MPT64 results on FNA | 35/37 (95%) | 14/15 (93%) | 15/16 (94%) | 3/19 (16%) |
| **Granulomatous inflammation +/- necrosis** | 18/19 (95%) | 8/8 (100%) | 8/9 (89%) | 0 |
| **Mixed inflammatory cells with necrosis** | 12/12 (100%) | 5/5 (100%) | 5/5 (100%) | 0 |
| Suppurative inflammation +/- necrosis | 4/4 (100%) | 1/1 (100%) | 1/1 (100%) | 1/1 (100%) |
| Reactive lymph node hyperplasia | 0/1 (0%) | 0/1 (0%) | 0 | 1/2 (50%) |
| Inconclusive | 1/1 (100%) | 0 | 1/1 (100%) | 0/2 (0%) |
| Malignancy | 0 | 0 | 0 | 0/7 (0%) |
| Benign tumor | 0 | 0 | 0 | 1/7 (14%) |
| | All EPTB (n = 38)1 | HIV+ (n = 17) | HIV- (n = 19) | Non-TB (n = 26) |
| Total positive MPT64 results in effusions (pleural/ascites) | 30/38 (79%) | 11/17 (65%) | 17/19 (90%) | 5/26 (19%) |
| **Epitheloid cells +/- mixed inflammatory cells and +/- necrosis** | 2/2 (100%) | 0 | 2/2 (100%) | 0 |
| **Lymphocytes +/- macrophages** | 17/24 (71%) | 6/12 (50%) | 10/11 (91%) | 3/12 (25%) |
| **Mixed inflammatory cells with necrosis** | 6/6 (100%) | 2/2 (100%) | 3/3 (100%) | 0 |
| Mixed inflammatory cells without necrosis | 2/3 (67%) | 1/1 (100%) | 1/2 (50%) | 2/5 (40%) |
| Inconclusive | 3/3 (100%) | 2/2 (100%) | 1/1 (100%) | 0/6 (0%) |
| Malignancy | 0 | 0 | 0 | 0/2 (0%) |
| Benign mesothelial cells | 0 | 0 | 0 | 0/1 (0%) |

1 Patients with unknown serostatus is included in this group.

had samples examined by both tests. The remaining TB case that would not have been identified most likely had the wrong organ sampled.

The cytology criteria defined to be compatible with TB are highlighted in bold in Table 4.

The MPT64 test was positive in 78% of all TB patients that had cytological findings compatible with TB, as shown in Table 4. There was 5 false positive MPT64 results; 1 pleural effusion and 4 ascites samples.

## Discussion

We have shown that the MPT64 antigen detection test for EPTB was implementable in Mbeya Zonal Referral Hospital, a tertiary level hospital in Tanzania. In a high TB/high HIV setting the test greatly improved the diagnosis of EPTB, irrespective of HIV status. The performance of the MPT64 test was not significantly reduced among PLWHIV, in contrast to both Mtb culture and GeneXpert that showed a significantly reduced sensitivity among PLWHIV.

The clinicians were asked to recruit patients with a clinical suspicion of EPTB and we were able to classify 56% as TB cases. Even in a high TB/high HIV setting 44% of suspected EPTB cases were not classified as having TB, but other non-TB diseases. This replicates results from our pediatric cohort in Mbeya and our study in Zanzibar and shows that differential diagnosis also needs to be considered in the management of presumptive EPTB [16,17]. In our study, a significantly higher percentage of non-TB patients (20% vs 7%, non-TB vs TB, respectively) had received anti-TB therapy in the past, which would imply underuse of appropriate diagnostic tools, and over-treatment with anti-TB therapy on clinical suspicion alone. Thus, potentially exposing patients to unnecessary side-effects and delayed diagnosis. This is supported by results in a major HIV registry study where only 438/765 (57.3%) EPTB patients had a confirmatory bacteriological test performed, and of those who had a test performed only 42% were positive [30].

In our cohort we found a very high HIV prevalence (46% vs 42%, TB vs non-TB, respectively). Few patients had CD4 counts available, but almost all (93% vs 90%) were reported to be on ART. A good ART coverage can also be reflected in the finding that the majority of patients had suspected TB adenitis, whilst a poor coverage would lead to more severe EPTB presentations such as disseminated TB or tuberculous meningitis [31]. Adenitis, pleuritis and peritonitis were the most common infections, and this was also found in the mentioned HIV registry study [30]. Only 9% of the TB patients had PTB with EPTB manifestations, which highlights the need to actively search for and sample extrapulmonary lesions to find the correct diagnosis.

Mtb culture is the gold standard test for infection with Mtb but is known to a have suboptimal sensitivity on extrapulmonary samples due to the low mycobacterial load [6,32]. Mycobacterial antigen accumulation and delayed hypersensitivity reaction to mycobacterial antigens has been suggested to play a central role in the pathogenesis and tissue destruction, even though the live bacterial load is low [33,34]. In this study we found an overall sensitivity of 28%, and a significantly poorer performance among PLWHIV. However, the results are mainly influenced by a poor sensitivity in effusions among HIV infected. The finding of a lower sensitivity in HIV infected is not supported by other studies [33,35–37]. Mtb culture had a comparable, but slightly lower sensitivity than GeneXpert. The results can be influenced by the number of contaminated culture samples, the transportation of the samples to the Central Tuberculosis Reference Laboratory in Dar es Salaam, and that GeneXpert detects DNA while culture detects live tubercle bacilli. Mtb culture is necessary for drug susceptibility testing, and measures such as improved logistical support and implementation of automated liquid culture systems should be taken to improve laboratory services. Turn-around time and poor sensitivity makes it unsuitable to serve as a stand-alone test for EPTB.

Previous reviews using extrapulmonary samples have found GeneXpert to have a comparable, but slightly lower sensitivity when compared to Mtb culture [9]. In our study we found a good correlation between positive GeneXpert and Mtb culture samples, but a slightly higher overall sensitivity of GeneXpert of 33%. As Mtb culture, GeneXpert had a significantly reduced sensitivity in PLWHIV which is explained by a poor performance in effusions. Few GeneXpert results were labelled as invalid or missing, and it is not likely that the higher sensitivity of GeneXpert in this study is due to false results. GeneXpert is specific, gives a rapid test result and offers resistance testing against Rifampicin. However, the sensitivity is too poor for the test to serve as a stand-alone test, cost when not subsidized is an issue, and electrical power, water supply and logistical issues have been shown to reduce its impact in real life [38,39]. The GeneXpert MTB/RIF Ultra is more sensitive than older versions for both PTB and EPTB, but reduced specificity is a concern [40].

The MPT64 test had a higher sensitivity than both Mtb culture and GeneXpert, whilst maintaining an acceptable specificity, confirming previous findings [10,16–18]. The test had a perfect correlation with true positive results on the mentioned tests and had a near perfect correlation with positive findings on FNAC. The test results were robust, and not significantly influenced by HIV serostatus, confirming the results from the pediatric cohort in Mbeya, where various causes of immunosuppression did not affect the test performance [17]. The good performance of the MPT64 test could be due to an increased presence of Mtb antigens rather than viable bacteria in EPTB lesions, and fits well with the theory of accumulation of Mtb antigens and their role in delayed type hypersensitivity reactions and disease pathogenesis [34]. Unfortunately, 27 test results were missing for the MPT64 test and 10 samples had an uncertain test result. These 37 samples could have affected the assessment of the test performance both ways. Quality control of the reading of the immunostained slides was performed by an outside pathologist, who reported ambiguity when evaluating the staining, however, this hesitancy was not reported by the local pathologist. This underlines that local variations in staining patterns affect the interpretation of the results, and that the pathologists need to be familiar with the staining. The test is dependent on existing pathology services and cannot be implemented until pathology services are established. However, with logistical support diagnostic sampling with FNA can be performed in a decentralized location, and cytology slides can be fixated and transported to a central laboratory for further analysis. The technique of immunostaining is also of great value in the diagnosis of cancers and would help strengthen health systems generally. Our results imply that the MPT64 test can diagnose EPTB with a high level of precision irrespective of HIV status in both high- and low resource settings.

Fine Needle Aspiration Cytology performed well and with the FNAC criteria for TB, showed a sensitivity of 86% on 44 samples, of which 39 were from lymph nodes. FNAC of solid lesions should be used more for the diagnosis of EPTB [7,41]. It is cheap, less invasive than biopsies and can be performed locally before transportation to a central pathology laboratory. The test would also give important information about non-TB diseases such as cancers [17]. However, the specificity is dependent on the context and prevalence of non-TB granulomatous inflammations could significantly reduce the specificity. Combining FNAC with MPT64 test showed excellent potential in this study with a near perfect correlation with positive FNAC results. A positive MPT64 results would increase the pathologists confidence with the TB diagnosis when considering other causes of granulomatous inflammation/necrosis, and a combination of FNAC and the MPT64 test would help identify TB cases without typical FNAC findings.

Cytology findings in bodily fluids is often unspecific [7]. Unfortunately, quantification of lymphocytes, adenosine deaminase activity, lactate dehydrogenase or of proteins in effusions were not available in this study. As cytology results were part of the composite reference

standard, caution was taken to make sure that no patient was classified as a TB case based on cytology findings alone.

Our study was a real-life study where the MPT64 test was implemented in a routine diagnostic setting. Studies on new diagnostic tools for EPTB is challenging due to Mtb culture being an imperfect gold standard that does not detect all cases. To mitigate this, we chose to validate the MPT64 test against a composite reference standard (CRS). The use of a CRS to classify patients resembles clinical practice, where the diagnosis of EPTB is often made by the integration of several non-specific clues from various investigations [42]. Response to anti-TB treatment was part of our CRS, which resembles clinical practice in the resource limited setting, where many patients are treated for presumptive EPTB. Response to treatment is, however, not uniform as especially Rifampicin has a wide spectrum and would treat many other conventional bacterial infections. Working with a CRS comes with a risk of misclassification bias and over/under assessment of the performance of the diagnostic test [43]. The tests in our CRS do not have a perfect specificity. We have tried to account for this by grading the certainty of our TB diagnosis to; confirmed TB, probable and possible. To reduce misclassification bias, an independent classification was done by TM if there remained doubt after the initial classification. The sample size in this study is fairly small which could reduce the strength of some of our findings. Patients were not tested for HIV as part of the study, which potentially could have reduced the number of people who were truly HIV-infected. A strength of this study was the access to the electronic hospital records, which gave comprehensive patient histories and supporting information.

## Conclusion

We have shown that the MPT64 test can diagnose EPTB with a high level of precision, regardless of HIV serostatus, thus outperforming currently used and WHO recommended tests. The test was implementable in a tertiary level hospital in Mbeya, Tanzania, and the local staff were able to use and apply the test in clinical practice. While treatment for TB is free of cost, the access to a diagnosis remains a major obstacle for patients with suspected EPTB as sampling of lesions is not free and is therefore performed infrequently. The implementation of an immunochemistry-based test is not feasible until a basic pathology service is in place. Strengthening health systems with appropriate diagnostic facilities should be a main priority for reaching universal health coverage.

## Supporting information

**S1 Table. Positive test results across different specimens among HIV+/HIV- study participants.** All TB/non-TB rows includes participants with unknown HIV status.
(XLSX)

**S2 Table. Correlation of cytology findings on FNA with positive results on different diagnostic test in all EPTB, HIV infected/ HIV uninfected EPTB and in non-TB patients.**
(XLSX)

**S3 Table. Correlation of cytological findings in effusions with postitive results on diagnostic tests in all EPTB, in HIV infected/uninfected EPTB and in non-TB.**
(XLSX)

**S1 Data. Datasets.**
(XLSX)

**S1 Text. Inclusivity in global research.**
(DOCX)

## Acknowledgments

We thank Prof. Harald G. Wiker for his efforts in developing the polyclonal antibody, Melissa Davidsen Jørstad for developing the questionnaires and routines used in the study, The *National Tuberculosis* and Leprosy Programme and the Central Tuberculosis Reference Laboratory at Muhimbili National Hospital in Dar es Salaam for supporting the study, the staff at Mbeya Zonal Referral Hospital for their dedicated recruitment, treatment and follow up of patients, Anthony Ambikile Nsojo in the Research department at Mbeya Zonal Referral Hospital for facilitating the study, the Histopathology unit at Mbeya Zonal Referral Hospital, pathologists Msafiri Marijani from Mnazi Mmoja Hospital and Innocent Mosha from Muhimbili National Hospital for their commitment to the patients and the study.

## Author Contributions

**Conceptualization:** Lisbet Sviland, Sayoki Mfinanga, Tehmina Mustafa.

**Data curation:** Erlend Grønningen, Marywinnie Nanyaro.

**Formal analysis:** Erlend Grønningen, Tehmina Mustafa.

**Funding acquisition:** Tehmina Mustafa.

**Investigation:** Erlend Grønningen, William Muller, Lisete Torres.

**Methodology:** Lisbet Sviland, Sayoki Mfinanga, Tehmina Mustafa.

**Project administration:** Esther Ngadaya, Sayoki Mfinanga, Tehmina Mustafa.

**Resources:** Esther Ngadaya, Sayoki Mfinanga, Tehmina Mustafa.

**Software:** Esther Ngadaya, Tehmina Mustafa.

**Supervision:** Lisbet Sviland, Esther Ngadaya, Sayoki Mfinanga, Tehmina Mustafa.

**Validation:** Marywinnie Nanyaro, Lisbet Sviland.

**Visualization:** William Muller, Lisete Torres.

**Writing – original draft:** Erlend Grønningen.

**Writing – review & editing:** Erlend Grønningen, Marywinnie Nanyaro, Lisbet Sviland, Esther Ngadaya, Sayoki Mfinanga, Tehmina Mustafa.

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
