## [Decision Letter · Decision Letter 0]

12 Sep 2022

PGPH-D-22-01191

Improved diagnosis of extrapulmonary tuberculosis in HIV infected adults in Mbeya, Tanzania using the MPT64 antigen detection test

Dear Dr.Grønningen,

Thank you for submitting your manuscript to PLOS Global Public Health. After careful consideration, we feel that it has merit but does not fully meet PLOS Global Public Health’s publication criteria as it currently stands. Therefore, we invite you to submit a revised version of the manuscript that addresses the points raised during the review process.

We look forward to receiving your revised manuscript.

Kind regards,

Jeannine Uwimana-Nicol, Ph.D.

Academic Editor

Journal Requirements:

1. Our staff editors have determined that your manuscript is likely within the scope of our Diagnostics in Global Health Call for Papers. This editorial initiative is headed by a team of Guest Editors for PLOS GPH: Senjuti Saha (Child Health Research Foundation, Bangladesh) and Titus Divala (Public Health Scotland, University of Glasgow and University of Malawi College of Medicine). The Collection will encompass a diverse range of research articles about diagnostics in global health, including innovation and deployment of point of care diagnostics; subsets of diagnostics related to infectious diseases, chronic diseases and injuries; policies related to and regulation of diagnostics; supply chain issues; and the affordability, accessibility, and availability of essential diagnostics..  Additional information can be found on our announcement page: https://collections.plos.org/call-for-papers/diagnostics-in-global-health/

If you would like your manuscript to be considered for this collection, please let us know in your cover letter and we will ensure that your paper is treated as if you were responding to this call.  Please note that being considered for the Collection does not require additional peer review beyond the journal’s standard process and will not delay the publication of your manuscript if it is accepted by PLOS GPH. If you would prefer to remove your manuscript from collection consideration, please specify this in the cover letter. 

2. Please include a complete copy of PLOS’ questionnaire on inclusivity in global research in your revised manuscript. Our policy for research in this area aims to improve transparency in the reporting of research performed outside of researchers’ own country or community. The policy applies to researchers who have travelled to a different country to conduct research, research with Indigenous populations or their lands, and research on cultural artefacts. The questionnaire can also be requested at the journal’s discretion for any other submissions, even if these conditions are not met.  Please find more information on the policy and a link to download a blank copy of the questionnaire here: https://journals.plos.org/globalpublichealth/s/best-practices-in-research-reporting. Please upload a completed version of your questionnaire as Supporting Information when you resubmit your manuscript.”

3. Please amend your detailed Financial Disclosure statement. This is published with the article. It must therefore be completed in full sentences and contain the exact wording you wish to be published.

a. State what role the funders took in the study. If the funders had no role in your study, please state: “The funders had no role in study design, data collection and analysis, decision to publish, or preparation of the manuscript.”

4. We do not publish any copyright or trademark symbols that usually accompany proprietary names, eg  ©, ®, ™  (e.g. next to drug or reagent names). Please remove all instances of trademark/copyright symbols throughout the text, including ® on page 35.

Additional Editor Comments (if provided):

Thank you for submitting your manuscript to PLOS Global Health, and we are glad to inform you that the peer review process is now complete.

The current status of the manuscript can not be accepted for publication as the manuscript requires Major revision.

Please attend to the reviewers' comments and submit for re-assessment in two weeks time.

Warm regards,

Dr Jeannine Uwimana Nicol

Academic Editor

Reviewers' comments:

Reviewer's Responses to Questions

**Comments to the Author**

1. Does this manuscript meet PLOS Global Public Health’s publication criteria? Is the manuscript technically sound, and do the data support the conclusions? The manuscript must describe methodologically and ethically rigorous research with conclusions that are appropriately drawn based on the data presented.

Reviewer #1: Yes

Reviewer #2: Yes

2. Has the statistical analysis been performed appropriately and rigorously?

Reviewer #1: Yes

Reviewer #2: Yes

3. Have the authors made all data underlying the findings in their manuscript fully available (please refer to the Data Availability Statement at the start of the manuscript PDF file)?

Reviewer #1: Yes

Reviewer #2: Yes

4. Is the manuscript presented in an intelligible fashion and written in standard English?

Reviewer #1: Yes

Reviewer #2: Yes

5. Review Comments to the Author

Reviewer #1: The study is interesting, well designed, well conducted. The results are robust and the manuscript well written. There are important clinical implications for the diagnosis extra-pulmonary TB in the HIV-infected individuals. It is excellent the rigor of the study conduction, especially considering the difficulties to run it in a low resource contry.

Few comments:

1. Please, change “HIV infected” as HIV-infected in the whole text

2. RESULTS: is any possibility to have at least a figure with the data represented in a graph? A colored graph is welcome. This will help the reader to easily understand the message of the high accuracy of the MPT64 antigen detection test for the diagnosis of extra-pulmonary TB

3. please, in the discussion, mention about the potential clinical implications of your findings for the diagnosis extra-pulmonary TB in the HIV-infected individuals vs HIV-uninfected in both, low and high resources countries.

4. please, mention the use of IGRA in the context of extra-pulmonary TB stating the limits. You may also consider to mention Goletti D et al. Int J Infect Dis. 2022 Mar 5:S1201-9712(22)00126-6. doi: 10.1016/j.ijid.2022.02.047; Petruccioli E, J Infect. 2020 May;80(5):536-546. doi: 10.1016/j.jinf.2020.02.009.

5. to

Reviewer #2: The authors investigated the implementation and performance of a novel diagnostic test for EPTB known as MPT64. The authors performed various cross-validations of the assay and concluded that it is implementable in a tertiary hospital setting in Tanzania. They also found that the MPT64 “performed better than currently used diagnostic tests”.

Major

The main issue is the results. I find it unnecessary to spend so much time discussing ZN staining results when it is already known that it performs poorly for EPTB. I suggest removing all results related to ZN throughout the results section. You even say in the discussion “ZN staining was of poor diagnostic value in this study”

I also find it strange that the authors spend so much time comparing the performance of culture and Xpert to the CRS and to each other. This paper is supposed to present findings related to MPT64, all these other results are distracting. I also don’t know what the sensitivity and specificity being reported is in comparison to for all these assays. Line 259: 28% sensitivity to what?

The word “correlation” to describe the relationships between assays in Table 4 is not the right word. These 2x2 comparisons are in fact describing the sensitivity/specificity relationships between assays.

My suggestion for the results is to keep the focus on MPT64. Remove all sens/spec/ppv/npv/accuracy for GeneXpert, culture, and ZN. Build 2x2 tables for all participants comparing it to 1) CRS; 2) Xpert; 3) culture. This will allow you to simultaneously compare the performance of MPT64 across different assays and draw conclusions on its performance. This means removing tables 2-4. These can be done for total pop, HIV-pos, HIV-neg. Supplementary can contain sample specific 2x2.

For tables 5 and 6, remove ZN, culture, Xpert. Shouldn’t table 5 be PLWHIV and table 6 HIV negative instead of TB positive and TB negative? The objective of the paper is to compare PLWHIV with HIV negative so combining these two groups doesn’t answer the research question. Either way, would consider merging these two tables so that the two groups (whatever they are) are side by side for easy comparison.

Minor

• Please use people living with HIV (PLWHIV) throughout instead of “HIV infected”.

• Line 1: Is this study truly focused on PLWHIV? The title makes it seem so, however more than half the study population were not living with HIV. I think the title needs to be clarified that this is a comparison the performance between the two populations.

• Line 30: Tanzania, not an “African setting”

• Line 36: general information about what the CRS was could be useful (test based, clinical based etc)

• Line 44-45: the long list of percentages is very difficult to read. Please put these immediately next to the statistic it represents to make it more clear “sensitivity (91% vs 78%)”

• Line 85-88: references are missing for these sentences

• Line 83-94: reference missing for AFB staining

• Line 97-102: is the info on LAM and ADA tests necessary? You never use these assays so it just feels like it extends the intro unnecessarily

• Line 127-128: Is there a reference for any published papers that used this cohort?

• Line 142: Self-reported HIV status could introduce some sort of bias, which could under estimate the number of people who are truly positive, which could be related to social desirability or recall. Might be worth mentioning in the limitations and suggesting that a more thorough approach would be to test participants for HIV at enrollment

• Line 150: you are missing a clear definition of the CRS in this section – what assays/components form the CRS. Even if it is presented in the figure I think it should be repeatd here. You should also explicitly state that the definition for each category is in figure 1 if you do not wish to define it explicitly in the text.

• Line 202: You talk of “agreement” as a form of quality control. How were these 20 slides selected? Perhaps you can quantify the level of agreement instead of reporting number of concordant/discordant pairs. Also, if LS labelled 7 inconclusive and the other examiner provided a result these are also part of the discordant pair, representing an agreement of 55%

• Table 1: Specify that “self reported previous TB” is treatment >1 year prior to enrollment. Need to include text under the table defining all the abbreviations like IPD, OPD, TB etc.

• The discussion section is unusually long. This needs to be greatly reduced to make it succinct.

6. PLOS authors have the option to publish the peer review history of their article (what does this mean?). If published, this will include your full peer review and any attached files.

**Do you want your identity to be public for this peer review?** For information about this choice, including consent withdrawal, please see our Privacy Policy.

Reviewer #1: No

Reviewer #2: No

---

## [Decision Letter · Decision Letter 1]

2 Nov 2022

Improved diagnosis of extrapulmonary tuberculosis in adults with and without HIV in Mbeya, Tanzania using the MPT64 antigen detection test

PGPH-D-22-01191R1

Dear Mr. Grønningen,

We are pleased to inform you that your manuscript 'Improved diagnosis of extrapulmonary tuberculosis in adults with and without HIV in Mbeya, Tanzania using the MPT64 antigen detection test' has been provisionally accepted for publication in PLOS Global Public Health.

Best regards,

Raquel Muñiz-Salazar, Ph.D.

Academic Editor

The authors answered the questions raised.

However, it is important to correct the following paragraph according to the reviewer comment:

Only a mistake in the text: serological tests are not the tests based on IFN-gamma detection. Then, I would suggest to modify the text at row 98 page 8 as:

"Serological tests and IFN-γ release assays lack accuracy to identify the progressors from infection to TB disease and are not suitable to detect EPTB (12)" and I would add the following reference for the serological tests:

Melkie ST,et al. The role of antibodies in tuberculosis diagnosis, prophylaxis and therapy: a review from the ESGMYC study group. Eur Respir Rev. 2022 Mar 9;31(163):210218. doi: 10.1183/16000617.0218-2021. PMID: 35264411

Reviewer Comments (if any, and for reference):

Reviewer's Responses to Questions

**Comments to the Author**

1. If the authors have adequately addressed your comments raised in a previous round of review and you feel that this manuscript is now acceptable for publication, you may indicate that here to bypass the “Comments to the Author” section, enter your conflict of interest statement in the “Confidential to Editor” section, and submit your "Accept" recommendation.

Reviewer #1: All comments have been addressed

Reviewer #2: All comments have been addressed

2. Does this manuscript meet PLOS Global Public Health’s publication criteria? Is the manuscript technically sound, and do the data support the conclusions? The manuscript must describe methodologically and ethically rigorous research with conclusions that are appropriately drawn based on the data presented.

Reviewer #1: Yes

Reviewer #2: Yes

3. Has the statistical analysis been performed appropriately and rigorously?

Reviewer #1: Yes

Reviewer #2: Yes

4. Have the authors made all data underlying the findings in their manuscript fully available (please refer to the Data Availability Statement at the start of the manuscript PDF file)?

Reviewer #1: Yes

Reviewer #2: (No Response)

5. Is the manuscript presented in an intelligible fashion and written in standard English?

Reviewer #1: Yes

Reviewer #2: Yes

6. Review Comments to the Author

Reviewer #1: the authors answered the questions raised.

Only a mistake in the text: serological tests are not the tests based on IFN-gamma detection. Then, I would suggest to modify the text at row 98 page 8 as:

"Serological tests and IFN-γ release assays lack accuracy to identify the progressors from infection to TB disease and are not suitable to detect EPTB (12)" and I would add the following reference for the serological tests:

Melkie ST,et al. The role of antibodies in tuberculosis diagnosis, prophylaxis and therapy: a review from the ESGMYC study group. Eur Respir Rev. 2022 Mar 9;31(163):210218. doi: 10.1183/16000617.0218-2021. PMID: 35264411

Reviewer #2: Thank you for the revision. The authors have sufficiently responded to my suggestions. The paper is much more succinct and focused on highlighting the important outcomes related to the MPT64 assay.

One small editorial suggestion: instead of "non-TB" I would just say "TB negative". For things like "TB samples", "TB positive samples" is more intuitive.

7. PLOS authors have the option to publish the peer review history of their article (what does this mean?). If published, this will include your full peer review and any attached files.

**Do you want your identity to be public for this peer review?** For information about this choice, including consent withdrawal, please see our Privacy Policy.

Reviewer #1: No

Reviewer #2: No
